# A Thirty-Minute Nap Enhances Performance in Running-Based Anaerobic Sprint Tests during and after Ramadan Observance

**DOI:** 10.3390/ijerph192214699

**Published:** 2022-11-09

**Authors:** Fatma Hilal Yagin, Özgür Eken, Ramazan Bayer, Vaclav Salcman, Tomasz Gabrys, Hürmüz Koç, Burak Yagin, İsmihan Eken

**Affiliations:** 1Department of Biostatistics and Medical Informatics, Faculty of Medicine, Inonu University, Malatya 44000, Turkey; 2Department of Physical Education and Sport Teaching, Inonu University, Malatya 44000, Turkey; 3Department of Gerontology, Malatya Turgut Ozal University, Malatya 44210, Turkey; 4Department of Physical Education and Sport, Faculty of Education, University of West Bohemia, 30100 Pilsen, Czech Republic; 5Department of Movement and Training Sciences, Faculty of Education, Canakkale Onsekiz Mart University, Canakkale 17020, Turkey

**Keywords:** nap, fasting, running-based anaerobic sprint test, kickboxing, health

## Abstract

The purpose of this study was to determine the impact of a 30 min nap (N30) on the Running-Based Anaerobic Sprint Test (RAST) both during and after Ramadan. Ten physically active kickboxers (age: 21.20 ± 1.61 years, height: 174.80 ± 4.34 cm, body mass: 73.30 ± 7.10 kg and body mass index (BMI): 24.00 ± 2.21 kg/m^2^) voluntarily performed the RAST test after an N30 and in a no-nap condition (NN) during two experimental periods: the last ten days of Ramadan (DR) and ∼3 weeks after Ramadan (AR). During each DR-NN, DR-N30, AR-NN and AR-N30 protocol, kickboxers performed RAST performance. A statistically significant difference was found between Ramadan periods (DR vs. AR) in terms of max power (W) (F = 80.93; *p*_1_ < 0.001; *η*^2^*_p_* = 0.89), minimum power (W) (F = 49.05; *p*_1_ < 0.001; *η*^2^*_p_* = 0.84), average power (W) (F = 83.79; *p*_1_ < 0.001; *η*^2^*_p_* = 0.90) and fatigue index (%) results (F = 11.25; *p*_1_ = 0.008; *η*^2^*_p_* = 0.55). In addition, the nap factor was statistically significant in terms of the max power (W) (F = 81.89; *p*_2_ < 0.001; *η*^2^*_p_* = 0.90), minimum power (W) (F = 80.37; *p*_2_ < 0.001; *η*^2^*_p_* = 0.89), average power (W) (F = 108.41; *p*_2_ < 0.001; *η*^2^*_p_* = 0.92) and fatigue index (%) results (F = 16.14; *p*_2_ = 0.003; *η*^2^*_p_* = 0.64). Taking a daytime nap benefits subsequent performance in RAST. The benefits of napping were greater after an N30 opportunity for DR and AR.

## 1. Introduction

During the month of Ramadan, Muslims who are in good health observe a fast that lasts roughly 29–30 days from sunrise to sunset. Depending on the season, fasting involves abstaining from food and drink for 13 to 18 h each day [1,2]. Every Muslim is allowed to eat and drink until sahour (dawn), after which they may fast until the time of the evening prayer (until sunset). Depending on the location’s geography and the season, the gap between night and early morning and the consequences that ensue can change significantly. Ramadan fasting is a special kind of fasting that is total (i.e., absolute avoidance of food as well as hydration), time-restricted, intermittent and circadian in nature. This makes it different from other fasting regimes such as calorie restriction (circadian rhythm and biological clock based on human rhythm). Thus, fasting athletes may experience hypohydration, changed sleep architecture and patterns, sleep disruptions, mood swings, immunological abnormalities, decreased psychomotor performance and generalized weariness that may be both physical and mental [3].

Studies on the impact of Ramadan on physical performance and behavior have found that, during the Wingate test, repeated sprint exercises, the 40 m sprint, agility tests, the 30 s repeated jump test and the multistage fitness test, specific physical performance problems are noted [4,5,6,7]. Changes in dietary and fluid intake, as well as sleep–wake patterns, may be the primary cause of decreased physical and cognitive performance [8,9,10]. It has also been reported that sleep efficiency, quality and duration decrease significantly during and after Ramadan (AR) compared to before the fasting month [11,12]. This may result in a significant increase in the duration of the light sleep phase during Ramadan (DR) as a result of the increased number of awakenings resulting from the increased nighttime metabolism caused by the late Souhour meal and the increase in foraging behaviors [13]. In addition, some of the most important issues for sports scientists and trainers are the stage of decision between fasting athletes DR, the risk of reducing their training load or continuing their usual training program in order to avoid negative training effects or stopping training [14]. It has been suggested for a long time that the changes brought on by intermittent fasting DR may necessitate a reduction in the training load undertaken by Muslim athletes [15]. A significant reduction in training load permits athletes to reduce the fatigue caused by intense training and enhance their performance in competition. It is crucial, therefore, to investigate whether athletes become more fatigued with DR and to take precautions in the event of a negative outcome [4].

Sleep is a significant component known to severely alter or hinder performance outcomes [16]. Athletes, who are frequently exposed to high-intensity training and competition programs due to the physiological and psychological restorative effects of sleep, may need more sleep than the general population due to increased mental and physical requirements [16]. Consequently, daytime napping has been utilized as a strategy to increase the quality and quantity of sleep in athletes [17]. It has been reported that napping is crucial for supporting nighttime sleep and optimizing physical performance, as the recommended 8 h of sleep per day may not be sufficient for athletes due to their physical and mental demands [16,18,19,20,21,22,23]. In addition, napping opportunity can positively affect short-term maximal performance, attention, feelings, muscle soreness, fatigue, stress ((no-nap opportunity (N0), 25 min of nap opportunity (N25)) [20], vigilance, shuttle run performance (N25) [24], 5 m Shuttle Run Test performance ((no-nap opportunity (N0), 25 min of nap opportunity (N25), a 35 min nap opportunity (N35), a 45 min nap opportunity (N45), a 90 min nap opportunity (N90)) [21,25], repeated sprint performance (a 20 min nap opportunity (N20), a 90 min nap opportunity (N90)) [26] and sprint performance [27].

When all of these studies in the literature were analyzed, it was determined that several of them investigated anaerobic performance. However, there are no studies evaluating kickboxers’ anaerobic performance following the opportunity to lie down. During kickboxing competitions, rounds are often finished within two minutes, and anaerobic energy methods are typically utilized. RAST measures anaerobic capacity and anaerobic power [28]. Because of this, kickboxers were used in our study, and it was also intended to bring something new to the body of knowledge. In addition, although further research is required to determine the impacts of various nap protocols on the performance of kickboxers, the period during and after Ramadan appears to be a significant aspect in determining the degree of such an effect. This design can aid coaches in identifying the ideal scenario and enhancing the competitive preparation of kickboxers via a nap. This study aimed to assess the effects of 30 min of napping (N30) and AR on kickboxers’ RAST performance.

## 2. Materials and Methods

### 2.1. Participants

The G*power software was utilized in order to determine the appropriate size of the sample [29,30]. As a result (alpha value = 0.05 and 1-beta value = 0.80, *η*^2^*_p_* = 0.25), it was reported that at least nine kickboxers should be included in the study [29]. Due to the risk of participants leaving, 16 kickboxers were chosen in all. Because they did not complete all of the required sessions, six of the kickboxers (n = 6) were disqualified from the data analysis. In this study, 10 physically active kickboxers participated. Their ages ranged from 21.20 ± 1.62, their height ranged from 174.80 ± 4.34 cm, their body mass ranged from 73.30 ± 7.15 kg and their body mass index ranged from 24.00 ± 2.21 kg/m^2^. Before beginning the study, the participants were provided with extensive information on its content, purpose and experimental design. Participants who volunteered to take part in the study signed a form indicating their agreement to the terms of the informed consent. Prior to the study, the participants reported that they intended to fast on the day of Ramadan when they were scheduled to study. In the post-Ramadan period, they were required to arrive full, having eaten at least three hours beforehand. All tests and evaluations used in this investigation were authorized by the Institute Clinical Research Ethics Committee (Approval Number: 2022/3534). During the phases of putting protocols into action and testing them following Ramadan, participants were reminded of the necessity of avoiding strenuous exercise and medications such as caffeine [31]. During the two data collection periods of this study, the average temperature and relative humidity were 18 °C and 58 percent DR and 20 °C and 52 percent AR, respectively. The inclusion criteria for the study participants were as follows: (a) doing kickboxing sports regularly (b) absence of a history of injury that could affect the results of the study; (c) adherence to the researchers’ instructions throughout the duration of the study; and (d) determination of the absence of any known sleep disorders. The exclusion criteria were (a) a history of sleep disorders; (b) disobedience to the investigators’ instructions during the study; and (c) the occurrence of any health issues during the performance tests.

### 2.2. Study Design

During the study, three familiarization sessions were carried out in order to ensure that the athletes participating in the study got used to the place where they would nap (napping) and to get them used to the RAST measurements after napping. After the familiarization sessions, the athletes came to the laboratory where the study would take place on the days and hours specified for the four protocols. According to findings from previous research, the effects of Ramadan on sprint performance may persist for at least two weeks after the end of Ramadan [6,32]. Due to this situation, the study was carried out in the middle of Ramadan (on the last 15 days) and two weeks after Ramadan in order to ensure the sufficient effect of Ramadan on the participants. The durations were established to be 30 min of no nap (NN) and nap opportunity DR and 30 min of NN and nap opportunity AR, respectively. At least 72 h were determined to occur between the nap and nap opportunities. After participants arrived at the laboratory, they were given ten minutes to adjust to their new sleeping environment. At 1:40 p.m., the participants were instructed to determine where they would lie down if they were given the opportunity. Beginning at 02:00 p.m., the participants were permitted to take an N30 in dark and quiet sleeping chambers. For DR and AR, participants in both the NN and N30 conditions engaged in routine tasks (such as using a cell phone and playing video games) without engaging in physical activity from the period following the nap opportunity until 4:50 p.m. At 4:50 p.m., they completed 5 min of a standardized warm-up consisting of two minutes of easy running followed by three minutes of specific exercises (i.e., foot sweeps, finger wrist and ankle rotations, trunk side stretch, trunk rotator stretch, hip circles, knee bends). At 5:00 p.m., after the warm-up, they performed an anaerobic sprint test based on running (RAST). At the conclusion of each experimental session, the participants were thanked, and at the conclusion of the entire experiment, they were properly debriefed and thanked for their participation.

### 2.3. Anthropometric Measurements

Volunteers (SECA^®^ Gmbh, Hamburg, Germany) were measured while standing and barefoot, with their ankles, calves, hips, scapula and head against the wall. The model of Frankfurt was accompanied by the head position, and height was measured upon the inhalation of air. When assessing body mass, participants wore light clothing (Toledo 2096 PP, So Bernardo do Campo, Brazil). The body mass index (BMI) was calculated by dividing weight (in kilograms) by height (m^2^) [33]. A Fitbit Charge 3 smart bracelet was utilized to assess if the athletes had entered the sleep phase during the 30 min sleep measurement. Fitbit devices use a microelectronic triaxial accelerometer to capture 3D body movement. This is a wearable wristband tracker that continuously measures and transmits data via Bluetooth to a smartphone or tablet using special algorithms to determine sleep. It is detected via Fitbit HR models by a proprietary light-emitting diode PurePulse^®^ technology. This photoplethysmography method consists of a light source and a photodetector that records changes in reflected light caused by changes in blood volume with each left ventricular contraction [34,35]. 

### 2.4. Running-Based Anaerobic Sprint Test (RAST)

The RAST test consisted of six 35 m maximal sprints separated by 10 s rest periods. A stopwatch was used after every effort to record the time (Casio HS-80TW, Shibuya, Tokyo, Japan). The power in Watts (W) for each sprint was determined by multiplying body mass (in kilograms) by the distance (35 m) raised to the second power. This result was then divided by the time of each sprint (T) raised to the third power in seconds (s). Maximum force equals body mass (kg) distance (m)^2^/time (s). The highest value of power produced during the sprints was referred to as peak power [28,33]. The highest number recorded is referred to as the RAST of the peak; the lowest number obtained is referred to as the RAST of minimum power; the sum of six repetitions divided by six is referred to as the RAST of average power; and the RAST of the fatigue index is derived from “Highest power − lowest power ÷ sum of time six sprints” [36]. The ICC of anaerobic power was 0.87, and the ICC of the fatigue index was previously reported to be 0.70 [37]. Additionally, the RAST test has been verified by earlier investigations [37,38].

### 2.5. Statistical Analysis

In this investigation, a two-way repeated-measures analysis of variance (interaction between Ramadan and nap) was employed. The assumption of normal distribution was examined using the Shapiro–Wilk test [39]. Mauchly’s sphericity test was performed for the sphericity assumption. A Greenhouse–Geisser correction for sphericity was used where necessary. Two groups (N30 and NN) were considered as the between-subject factor (group), and two measurements (DR and AR) were considered as the within-subject factor. These analyses were made for max power (W), minimum power (W), average power (W) and fatigue index (%) measurements. The findings are given using the mean value along with the standard deviation. Partial eta squared (*η*^2^*_p_*) was calculated for the effect size. The threshold for significance was set at *p* < 0.05. American Psychological Association (APA) 6.0 style was used to report statistical differences [40]. Python version 3.9 and IBM SPSS Statistics for Windows version 26.0 were utilized throughout the analysis process (New York, NY, USA).

## 3. Results

Table 1 shows the changes in the maximum power (W) parameter of the participants. According to Table 1, during the Ramadan period (DR vs. AR), the main effect was statistically significant, and the performance was significantly higher in AR (F = 80.93; *p*_1_ < 0.001; *η*^2^*_p_*= 0.89). Furthermore, there was a major effect for naps increasing maximum power (W) performance in both DR and AR (F = 81.89; *p*_2_ < 0.001; *η*^2^*_p_* = 0.90). The interaction effect (Ramadan period*nap) was statistically significant for maximum power (W) (F = 11.32; *p* = 0.008; *η*^2^*_p_* = 0.55).

The changes in the minimum power (W) parameter of the participants are presented in Table 2. According to the findings of the research, the main effect of Ramadan periods (DR vs. AR) (F = 49.05; *p*_1_ < 0.001; *η*^2^*_p_* = 0.84) and nap (F = 80.37; *p*_2_ < 0.001; *η*^2^*_p_* = 0.89) was statistically significant for minimum power (W) performance. Napping improved the performance in both DR and AR, although the minimum power (W) performance was lower in AR compared to DR. The interaction effect (Ramadan period*nap) was not statistically significant for minimum power (W) performance (F = 0.87; *p* = 0.37; *η*^2^*_p_* = 0.08).

The average power (W) performance differed between Ramadan periods and was higher in AR than in DR, as expected (F = 83.79; *p*_1_ < 0.001; *η*^2^*_p_* = 0.90). Moreover, the main effect of napping was statistically significant, and napping increased average power (W) performance (F = 108.41; *p*_2_ < 0.001; *η*^2^*_p_* = 0.92). There was an interaction effect (Ramadan period*nap) for mean power (W), and the results showed that napping combined with AR maximized performance (F = 6.08; *p* = 0.03; *η*^2^*_p_* = 0.40) (Table 3).

The Ramadan*nap interaction effect (F = 3.08; *p* = 0.11; *η*^2^*_p_* =0.25) was not significant for the fatigue index (%), but the main effect of Ramadan (F = 11.25; *p*_1_ = 0.008; *η*^2^*_p_* = 0.55) was significant and increased significantly in AR. Moreover, one-way further analyses showed that napping increased the fatigue index (%) (F = 16.14; *p*_2_ = 0.003; *η*^2^*_p_* = 0.64) (Table 4).

## 4. Discussion

The aim of this study is to investigate the effect of N30 opportunity on RAST performance during and after Ramadan in kickboxers. According to the results of the research, the maximum power (W) of the participants DR was significantly lower than it was after Ramadan. In addition, the maximum power (W) value was higher in the group that did N30 compared to the group that did not do N30. The Ramadan*nap interaction was statistically significant for the maximum power (W) value. The interaction results showed that performing N30 exercise AR will positively affect the maximum power (W) value. Participants’ minimum power (W) DR was significantly lower than it was after Ramadan. It can also be said that N30 exercises increase the minimum power (W) value. However, the Ramadan*nap interaction was not significant for minimum power (W). The average power (W) value DR was also significantly lower than AR. Additionally, the N30 increased the average power (W) of the participants. The Ramadan*nap interaction was significant for the average power (W) value. This interaction effect showed that exercise AR and N30 would increase the average power (W) value. The fatigue index (%) of the participants DR was also significantly lower than it was after Ramadan. In addition, the fatigue index (%) value was higher in the group that did N30 than it was for the group that did not do N30. The Ramadan*nap interaction was not significant for the fatigue index (%).

However, there are studies determining the effect of napping opportunity during Ramadan on different performance values of athletes. Hsouna et al. (2020) investigated the effects of a 35 min nap on physical and cognitive performance in physically active men before, during and after Ramadan. During the 5 m shuttle run test (5mSRT), neither total distance (TD) nor fatigue index (FI) were affected by Ramadan observance. However, the best distance (BD) was considerably lower in the last 10 days of Ramadan compared to 20 days AR in the absence of sleep (N0) and after a 35 min nap (N35). These authors hypothesized that further improvements in nap duration DR would allow for significant nap impacts on all parameters before, during and after the month of fasting [41]. Hsouna et al. (2020) investigated the effects of a 25 min nap opportunity on physical performance during the 5mSRT, feelings (measured by the feeling scale), attention (measured by the digit cancellation test) and the perception of fatigue (measured by the rating of perceived exertion (RPE)) during the observance of Ramadan. They revealed that a 25 min nap opportunity improved physical and cognitive performance after Ramadan observance; however, this nap length was insufficient to produce significant differences during the month of Ramadan [42]. Souissi et al. (2020) examined the effects of partial sleep deprivation (SDN) and an N30 on physical and cognitive performance and mood states. In conclusion, the results of the study demonstrated that an N30 opportunity improved mood states and cognitive and physical performance after a night of sleep deprivation and after a night of normal sleep. However, physiological parameters did not change significantly in response to exercise [43]. Boukhris et al. (2022) examined the effect of nap duration on short-duration repetitive maximal exercise performance and the perception of effort DR. The results of this study indicate that napping improves physical performance and perceived exertion before, during and after Ramadan. During the pre- and post-Ramadan testing periods, nap duration had little impact on the magnitude of the improvements induced by afternoon naps. Nevertheless, DR (the ER testing period), 45 min naps were more effective for enhancing performance and decreasing RPE scores during the 5mSRT [32]. Contrary to these studies, Hsouna et al. (2019) evaluated variations in the short-term maximal performance, alertness, dietary intake, sleep pattern and mood states of physically active young men before (BR), during and after the observance of Ramadan. The researchers discovered that Ramadan had no negative impact on the five-jump performance, alertness or mood states of physically active young men [44]. Hsouna et al. (2019) examined the impact of different nap opportunity durations (no nap opportunity (N0), 25 min of nap opportunity (N25), 35 min of nap opportunity (N35) and 45 min of nap opportunity (N45)) on short-term maximal performance, attention, feelings, muscle soreness, fatigue, stress and sleep. Compared to N0, they revealed that the five-jump performance improved significantly during N35 and N45. In addition, N45 improved concentration compared to N0. In comparison to N0, fatigue, sleep and stress levels were significantly reduced after N25, N35 and N45 [20]. Boukhris et al. (2019) examined the effect of daytime nap duration on repeated high-intensity short-duration performance and the evaluation of perceived effort (RPE). They determined that the best distance (BD) increased after N25 and N45 compared to N0, and it was substantially greater after N45 than it was after N35. Compared to N0, the three-nap duration improved total distance (TD), with the greatest improvement occurring after N45. There were no significant variations in fatigue index (FI) across the three nap opportunity lengths and N0. The mean RPE score after N25 and N0 was substantially greater than that after N45 [21]. Abdessalem et al. (2019) examined the effect of a daytime napping opportunity taken at various times of the day on physical and cognitive performance. The total distance (TD) at 17h00 was greater in the 14h00 nap condition than it was in the 13h00 nap condition or in the no-nap condition. After a nap at 14h00, HD was greater than it was in the no-nap condition, and after a nap at 15h00, HD was greater than it was in the no-nap condition. In addition, HD was greater after a nap at 14h00 and 15h00 than it was after a nap at 13h00. There was no effect of napping at 13:00 on physical performance at 17:00 [24]. Hammouda et al. (2018) examined the effects of 20 min (N20) and 90 min (N90) naps following partial sleep deprivation (PSD) on the hematological and biochemical responses to repeated-sprint exercise. They showed that all nap options improved repeated-sprint performance, although N90 resulted in a greater improvement. N90 resulted in increased post-exercise levels of Monocytes (MO), Lymphocytes (LY), Hemoglobin (HB), Hematocrit (HT), Sodium and Potassium [26].

When all these studies in the literature were examined, it was seen that there were some studies examining anaerobic performances. However, there is no study evaluating anaerobic performance after the opportunity to lie down in kickboxers. Rounds in kickboxing tournaments are completed within two minutes, and anaerobic energy systems are often used during competition. RAST assesses anaerobic power and anaerobic capacity [37]. Because of this situation, kickboxers were used in our study, and it was also aimed to add novelty to the literature. There are some limitations regarding the study. Kickboxers were asked to take food products equivalent to the number of calories they received during Ramadan after Ramadan. However, the differences between the diet (daily calorie intake) during the month of Ramadan and the diet after the month of Ramadan could not be completely controlled because the athletes were not in the camp environment. The findings of this study are subject to certain restrictions. In this particular study, the RAST performance at 5:00 p.m. was not compared to that of any other afternoon hour (1:00 p.m., 3:00. p.m.). Further, there was no assessment before Ramadan.

## 5. Conclusions

The findings of the current investigation revealed that taking a short nap improves RAST performance during and after the holy month of Ramadan. Throughout the testing periods that took place during and after Ramadan, the length of the afternoon naps had very little impact on the magnitude of the benefits that were observed. Data from the present study suggest that athletes should aim to nap for 30 min at 2:00 p.m. during and after Ramadan to improve physical performance. It has been reported that sleeping at 2:00 p.m. has an effect on one’s physical performance three hours later. It is possible to conduct another study similar to this one by expanding the size of the sample size taken from male and female athletes of varying ages. It has been stated that taking a nap for thirty minutes around 2:00 p.m. during and after the holy month of Ramadan will increase RAST performance, and this condition is one that coaches can recommend to their athletes.

## Figures and Tables

**Table 1 ijerph-19-14699-t001:** Comparison of the measured values of the Max Power (W) (n = 10).

Groups	Mean ± SD	Ramadan Period Main Effect	Nap Main Effect	Interaction
		F Value	F Value	
		*p*_1_ Value	*p*_2_ Value	
		*η* ^2^ * _p_ *	*η* ^2^ * _p_ *	
Max Power (W) DR-NN	680.90 ± 184.14	F = 80.93 *p*_1_ < 0.001 *η*^2^*_p_* = 0.89	F = 81.89 *p*_2_ < 0.001 *η*^2^*_p_* = 0.90	F = 11.32 *p* = 0.008 *η*^2^*_p_* = 0.55
Max Power (W) DR-N30	732.30 ± 186.71			
Max Power (W) AR-NN	703.50 ± 185.53			
Max Power (W) AR-N30	780.30 ± 200.64			

DR: during Ramadan, AR: after Ramadan, N30: 30 min nap, NN: no nap, Mean: mean values, SD: standard deviation, Ramadan period: (DR vs. AR), *p*_1_ Value: significance test result between Ramadan periods, *p*_2_ Value: significance test result between N30 and NN.

**Table 2 ijerph-19-14699-t002:** Comparison of the measured values of the Minimum Power (W) (n = 10).

Groups	Mean ± SD	Ramadan Period Main Effect	Nap Main Effect	Interaction
		F Value	F Value	
		*p*_1_ Value	*p*_2_ Value	
		*η* ^2^ * _p_ *	*η* ^2^ * _p_ *	
Minimum Power (W) DR-NN	363.60 ± 69.50	F = 49.05 *p*_1_ < 0.001 *η*^2^*_p_* = 0.84	F = 80.37 *p*_2_ < 0.001 *η*^2^*_p_* = 0.89	F = 0.87 *p* = 0.37 *η*^2^*_p_* = 0.08
Minimum Power (W) DR-N30	406.50 ± 86.30			
Minimum Power (W) AR-NN	381.80 ± 76.92			
Minimum Power (W) AR-N30	432.30 ± 75.57			

DR: during Ramadan, AR: after Ramadan, N30: 30 min nap, NN: no nap, Mean: mean values, SD: standard deviation, Ramadan period: (DR vs. AR), *p*_1_ Value: significance test result between Ramadan periods, *p*_2_ Value: significance test result between N30 and NN.

**Table 3 ijerph-19-14699-t003:** Comparison of the measured values of the Average Power (W) (n = 10).

Groups	Mean ± SD	Ramadan Period Main Effect	Nap Main Effect	Interaction
		F Value	F Value	
		*p*_1_ Value	*p*_2_ Value	
		*η* ^2^ * _p_ *	*η* ^2^ * _p_ *	
Average Power (W) DR-NN	500.90 ± 86.05	F = 83.79 *p*_1_ < 0.001 *η*^2^*_p_* = 0.90	F = 108.41 *p*_2_ < 0.001 *η*^2^*_p_* = 0.92	F = 6.08 *p* = 0.03 *η*^2^*_p_* = 0.40
Average Power (W) DR-N30	553.90 ± 94.99			
Average Power (W) AR-NN	523.80 ± 88.97			
Average Power (W) AR-N30	587.40 ± 103.28			

DR: during Ramadan, AR: after Ramadan, N30: 30 min nap, NN: no nap, Mean: mean values, SD: standard deviation, Ramadan period: (DR vs. AR), *p*_1_ Value: significance test result between Ramadan periods, *p*_2_ Value: significance test result between N30 and NN.

**Table 4 ijerph-19-14699-t004:** Comparison of the measured values of the Fatigue Index (%) (n = 10).

Groups	Mean ± SD	Ramadan Period Main Effect	Nap Main Effect	Interaction
		F Value	F Value	
		*p*_1_ Value	*p*_2_ Value	
		*η* ^2^ * _p_ *	*η* ^2^ * _p_ *	
Fatigue Index (%) DR-NN	9.41 ± 5.36	F = 11.25 *p*_1_ = 0.008 *η*^2^*_p_* = 0.55	F = 16.14 *p*_2_ = 0.003 *η*^2^*_p_* = 0.64	F = 3.08 *p* = 0.11 *η*^2^*_p_* = 0.25
Fatigue Index (%) DR-N30	10.10 ± 5.23			
Fatigue Index (%) AR-NN	9.67 ± 5.35			
Fatigue Index (%) AR-N30	10.92 ± 5.64			

DR: during Ramadan, AR: after Ramadan, N30: 30 min nap, NN: no nap, Mean: mean values, SD: standard deviation, Ramadan period: (DR vs. AR), *p*_1_ Value: significance test result between Ramadan periods, *p*_2_ Value: significance test result between N30 and NN.

## Data Availability

Data are available for research purposes upon reasonable request to the corresponding author.

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
