# Peer review of "A Thirty-Minute Nap Enhances Performance in Running-Based Anaerobic Sprint Tests during and after Ramadan Observance"

_ijerph, 2022, doi:10.3390/ijerph192214699_

Round 1
Reviewer 1 Report
The aim of the present investigation was to study the effects of a 30-min nap (N30) on the running-based anaerobic sprint test (RAST) during and after Ramadan among kickboxers. The results of the study showed that maximum, minimum and average power were significantly greater after Ramadan, and also in the N30 condition. Further, it was found a statistically significant interaction for Ramadan*NAP regarding maximum and average power, but not for minimum power. The authors concluded that performing a 30-min nap after Ramadan would increase RAST performance in kickboxers.
Hereunder, I provide some comments and suggestions for helping authors to improve the quality of their manuscript.
Please revise English language and use the SI along the whole manuscript.
Abstract:
Lines 16: Please write RAST abbreviation before “…both during and after Ramadan.”
Line 18: “30-min nap (N30)” is already abbreviated in line 15.
Lines 20-26: Please do not repeat the word “results”. It is redundant.
Line27: Use N30 abbreviation: “30-min nap” is already abbreviated in line 15.
You stated you found significant differences for many performance variables when comparing Ramadan periods (i.e., DR vs. AR), but you do not report in your abstract section in what condition was performance improved. Please modify this accordingly.
Introduction:
Line 62: Please change “…their usual training load…” for “…their usual training program…”.
I suggest authors to expand more on the effects of napping rather than reducing training load on subsequent exercise performance during and after Ramadan. You should provide a more solid rationale regarding the pertinence and main goal of your study.
Materials and methods:
Participants subsection:
Please provide specific inclusion/exclusion criteria.
Procedures subsection:
Lines 101-102: Please rewrite. Difficult to follow.
I have some concerns regarding procedures that are crucial for guarantying the replicability of the present study and for supporting the conclusions.
What did participants do in the non-nap condition during the 30 min? Please report this information for guarantying replicability.
How did you control if your participants really slept in the N30 condition? Did you measure the sleep quality during the 30-min nap? This is an important point since it affects the subsequent exercise performance in the RAST.
Please explain in detail how did you obtained the Fatigue Index values.
Why did you select the RAST for assessing exercise performance in kickboxers? Please provide a rationale of the appropriateness of this test (i.e., running-based) for kickboxers regarding sport specificity. You can also add a brief paragraph concerning this issue in the discussion section.
You did not report information about dietary patterns during and after Ramadan. In this regard, you did not control a variable (i.e., energy intake) that may play a key role in the improvements assessed after Ramadan. I suggest authors to reflect this in the discussion section as a limitation of your study.
Results:
You have already reported in the “Participants” subsection the demographic data. Furthermore, avoid to present duplicated information (e.g., demographic data in text-format and in Figure). Please modify it accordingly along the Results section.
Discussion:
Please be consistent with the use of abbreviations along the manuscript (e.g., N30), especially focus on the discussion section.
Line 230: Please change: “They hypothesized…” for “These authors hypothesized…”.
Authors are encouraged to expand the discussion section adding new information and focusing on the sleep quality in N30 condition, appropriateness of implementing the RAST for assessing exercise performance in kickboxers, fatigue index, and dietary patterns (please see previous comments).
Conclusions:
Line 262: What do you mean with "the present review"?
Author Response
Thank you for your comments about the study.
Comments & Corrections
Reviewer 1 Comments (revised changes shown as yellow highlight)
Comment 1: “Please revise English language and use the SI along the whole manuscript.”
Correction 1: Thank you. The required revision was made to the manuscript.
Comment 2: ‘‘Abstract, Lines 16: Please write RAST abbreviation before “…both during and after Ramadan.”’’
Correction 2: Thank you. The required revision was made to the manuscript.
Comment 3: ‘‘Abstract, Line 18: “30-min nap (N30)” is already abbreviated in line 15.’’
Correction 3: Thank you. I revised the abbreviations in the manuscript.
Comment 4: ‘‘Abstract, Lines 20-26: Please do not repeat the word “results.” It is redundant.’’
Correction 4: Thank you. The required revision was made to the manuscript.
Comment 5: ‘‘Abstract, Line27: Use N30 abbreviation: “30-min nap” is already abbreviated in line 15.’’”
Correction 5: Thank you. I revised the abbreviations in the manuscript.
Comment 6: ‘‘You stated you found significant differences for many performance variables when comparing Ramadan periods (i.e., DR vs. AR), but you do not report in your abstract section in what condition was performance improved. Please modify this accordingly. ’’
Correction 6: ‘‘ During each DR-NN, DR-N30, AR-NN and AR-N30 protocol, kickboxers performed RAST performance. ’’
Comment 7: ‘‘Introduction, Line 62: Please change “…their usual training load…” for “…their usual training program…”
Correction 7: Thank you. The required revision was made to the manuscript.
Comment 8: ‘‘Introduction, I suggest authors to expand more on the effects of napping rather than reducing training load on subsequent exercise performance during and after Ramadan. You should provide a more solid rationale regarding the pertinence and main goal of your study. ’’
Correction 8: Revised. ‘Sleep is a significant component known to severely alter or hinder performance outcomes [16]. Athletes, who are frequently exposed to high-intensity training and competition programs due to the physiological and psychological restorative effects of sleep, may need more sleep than the general population due to increased mental and physical requirements [16]. Consequently, daytime napping has been utilised as a strategy to increase the quality and quantity of sleep in athletes [17]. It has been reported that napping are crucial for supporting nighttime sleep and optimizing physical performance, as the recommended 8 hours of sleep per day may not be sufficient for athletes due to their physical and mental demands [16,18–23]. In addition napping opportunity can positively affect short-term maximal performance, attention, feelings, muscle soreness, fatigue, stress ((no-nap opportunity (N0), 25 min of nap opportunity (N25)) [20], vigilance, shuttle run (N25) [24], 5-m Shuttle Run Test ((no-nap opportunity (N0), 25 min of nap opportunity (N25), a 35 min nap opportunity (N35), a 45 min nap opportunity (N45)) (a 45 min nap opportunity (N45), a 90 min nap opportunity (N90)) [21,25], repeated sprint (a 20 min nap opportunity (N20), a 90 min nap opportunity (N90)) [26], sprint performance [27].’
Comment 9: “Materials and methods, Participants subsection:”
Correction 9: ‘‘Due to the risk of participants leaving, 16 kickboxers were chosen in all. Because they did not complete all of the required sessions, six of the kickboxers (n = 6) were disqualified from the data analysis. In this study, 10 physically active kickboxers participated.’’
Comment 10: ‘‘Materials and methods, Please provide specific inclusion/exclusion criteria.”
Correction 10: ‘‘The inclusion criteria for the study participants were as follows: (a) doing kickboxing sports regularly (b) absence of a history of injury that could affect the results of the study; (c) adherence to the researchers' instructions throughout the duration of the study; and (e) determination of the absence of any known sleep disorders. Exclusion criteria were (a) a history of sleep disorders; (b) disobedience to the investigators' instructions during the study; and (c) the occurrence of any health issues during the performance tests.’’
Comment 11: ‘‘Materials and methods, Procedures subsection:’’
Correction 11: I didn’t understand well. According to understand I changed ‘Procedures subsection’ as ‘Study Design’.
Comment 12: ‘‘Materials and methods, Lines 101-102: Please rewrite. Difficult to follow.‘’
Correction 12: ‘‘During the study, three familization sessions were carried out in order to ensure that the athletes participating in the study got used to the place where they would nap (napping) and to get them used to the RAST measurements after nap (napping). After the familization sessions, the athletes came to the laboratory where the study would take place on the days and hours specified for the four protocols. According to findings from previous research, the effects of Ramadan on sprint performance may persist for at least two weeks after the end of Ramadan [6,32]. Due to this situation, the study was carried out in the middle of Ramadan (on the last 15 days) and two weeks after Ramadan in order to ensure the sufficient effect of Ramadan in the participants.’’
Comment 13: ‘‘Materials and methods, What did participants do in the non-nap condition during the 30 min? Please report this information for guarantying replicability.’’
Correction 13: ‘‘ DR and AR, participants in both the NN and N30 conditions engaged in routine tasks (such as using a cell phone, playing video games) without engaging in physical activity from the period following the nap opportunity until 04:50 p.m. ’’
Comment 14: ‘‘Materials and methods, how did you control if your participants really slept in the N30 condition? Did you measure the sleep quality during the 30-min nap? This is an important point since it affects the subsequent exercise performance in the RAST.’’
Correction 14: ‘‘A Fitbit Charge 3 smart bracelet was utilised to assess if the athletes had entered the sleep phase during the 30-minute sleep measurement. Fitbit devices use a microelectronic triaxial accelerometer to capture 3D body movement; This motion data is a wearable wristband tracker that continuously measures and transmits data via Bluetooth to a smartphone or tablet using special algorithms to determine sleep. It is detected by Fitbit HR models by a proprietary light-emitting diode PurePulse® technology. This photoplethysmography method consists of a light source and a photodetector that records changes in reflected light caused by changes in blood volume with each left ventricular contraction [34,35]. ’’
Comment 15: ‘‘Materials and methods, Please explain in detail how did you obtained the Fatigue Index values.’’
Correction 15: The highest number recorded is referred to as the RAST of peak; the lowest number obtained is referred to as the RAST of minimum power; the sum of six repetitions divided by six is referred to as the RAST of average power; and the RAST of Fatigue Index is derived from "Highest power - lowest power ÷ sum of time 6 sprints" [36]. The ICC of anaerobic power was 0.87, and the ICC of fatigue index was previously reported to be 0.70 [37]. Additionally, the RAST test has been verified by earlier investigations [37,38].
Comment 16: ‘‘Materials and methods, Why did you select the RAST for assessing exercise performance in kickboxers? Please provide a rationale of the appropriateness of this test (i.e., running-based) for kickboxers regarding sport specificity. You can also add a brief paragraph concerning this issue in the discussion section.”
Correction 16: ‘‘When all these studies in the literature were examined, it was seen that there were some studies examining anaerobic performances. However, there is no study evaluating anaerobic performance after the opportunity to lie down in kickboxers. Rounds in kickboxing tournaments are completed within two minutes, and anaerobic energy systems are often used during competition. RAST assesses anaerobic power and anaerobic capacity. Because of this situation, kickboxers were used in our study and it was also aimed to add a novelty to the literature.’’
Comment 17: ‘‘Materials and methods, You did not report information about dietary patterns during and after Ramadan. In this regard, you did not control a variable (i.e., energy intake) that may play a key role in the improvements assessed after Ramadan. I suggest authors to reflect this in the discussion section as a limitation of your study.”
Correction 17: ‘‘There are some limitations regarding the study. Kickboxers were asked to take food products equivalent to the amount of calories they received during Ramadan after Ramadan. However, the differences in the diet (daily calorie intake) during the month of Ramadan and the diet after the month of Ramadan could not be completely controlled because the athletes were not in the camp environment.’’
Comment 18: ‘‘Results, You have already reported in the “Participants” subsection the demographic data. Furthermore, avoid to present duplicated information (e.g., demographic data in text-format and in Figure). Please modify it accordingly along the Results section.”
Correction 18: Thank you. Figure 1 has been removed from the manuscript.
Comment 19: “Discussion, Please be consistent with the use of abbreviations along the manuscript (e.g., N30), especially focus on the discussion section.”
Correction 19: Thank you. I revised the abbreviations in the manuscript.
Comment 20: “Discussion, Line 230: Please change: “They hypothesized…” for “These authors hypothesized…”.”
Correction 20: Thank you. The required revision was made to the manuscript.
Comment 21: “Discussion, Authors are encouraged to expand the discussion section adding new information and focusing on the sleep quality in N30 condition, appropriateness of implementing the RAST for assessing exercise performance in kickboxers, fatigue index, and dietary patterns (please see previous comments).”
Correction 21: ‘ Hsouna et al. (2019) examined the impact of different nap opportunity durations (no-nap opportunity (N0), 25 minutes of nap opportunity (N25), 35 minutes of nap opportunity (N35), and 45 minutes of nap opportunity (N45)) on short-term maximal performance, attention, feelings, muscle soreness, fatigue, stress, and sleep. Compared to N0, they revealed that the 5-jump performance improved significantly during N35 and N45. In addition, N45 improved concentration compared to N0. In comparison to N0, fatigue, sleep, and stress levels were significantly reduced after N25, N35, and N45 [20]. Boukhris et al. (2019) examined the effect of daytime nap duration on repeated high-intensity short-duration performance and evaluation of perceived effort (RPE). They determined that best distance (BD) increased after N25 and N45 compared to N0, and was substantially greater after N45 than N35. Compared to N0, the three-nap duration improved total distance (TD), with the greatest improvement occurring after N45. There were no significant variations in fatigue index (FI) across the three nap opportunity lengths and N0. The mean RPE score after N25 and N0 was substantially greater than after N45 [45]. Abdessalem et al. (2019) examined the effect of a daytime napping opportunity taken at various times of the day on physical and cognitive performance. Total distance (TD) at 17h00 was greater in the 14h00 nap condition than in the 13h00 nap condition or in the no-nap condition. After a nap at 14h00, HD was greater than in the no-nap condition, and after a nap at 15h00, HD was greater than in the no-nap condition. In addition, HD was greater after a nap at 14h00 and 15h00 than it was after a nap at 13h00. There was no effect of napping at 13:00 on physical performance at 17:00 [24]. Hammouda et al. (2018) examined the effects of 20-min (N20) and 90-min (N90) naps following partial sleep deprivation (PSD) on the haematological and biochemical responses to repeated-sprint exercise. They showed that all nap options improved repeated-sprint performance, although N90 resulted in a greater improvement. N90 resulted in increased post-exercise levels of Monocytes (MO), Lymphocytes (LY), Hemoglobin (HB), Hematocrit (HT), Sodium, and Potassium [26].
When all these studies in the literature were examined, it was seen that there were some studies examining anaerobic performances. However, there is no study evaluating anaerobic performance after the opportunity to lie down in kickboxers. Rounds in kickboxing tournaments are completed within two minutes, and anaerobic energy systems are often used during competition. RAST assesses anaerobic power and anaerobic capacity [37]. Because of this situation, kickboxers were used in our study and it was also aimed to add a novelty to the literature. There are some limitations regarding the study. Kickboxers were asked to take food products equivalent to the amount of calories they received during Ramadan after Ramadan. However, the differences in the diet (daily calorie intake) during the month of Ramadan and the diet after the month of Ramadan could not be completely controlled because the athletes were not in the camp environment. The findings of this study are subject to certain restrictions. In this particular study, the RAST performance at 5:00 p.m. was not compared to that of any other afternoon hour (01.00 p.m., 03.00. p.m.). Besides there was no assessment of before Ramadan.’
Comment 22: “Conclusions, Line 262: What do you mean with "the present review”?”
Correction 22: Thank you. It was changed to “the present study” in the manuscript.

Reviewer 2 Report
This study is an interesting paper that attempts to verify the impact of the sprint test before and after Ramadan on performance. I think that there are the following points that should be improved.
1. The originality of this study was not properly mentioned in the introduction. A clear description of questions such as why a 30-minute nap, why a RAST performance, and why a kickboxer should be added, and the uniqueness of this study should also be clearly described.
2.IJERPH requires pre-registration for clinical trial registration. The presence or absence of clinical trial registration should be stated.
3. I don't think Figure 1 is information that should be included in the diagram.
Author Response
Thank you for your valuable reviews.
Comments & Corrections
Reviewer 2 Comments (revised changes shown as green highlight)
Comment 1: ‘‘The originality of this study was not properly mentioned in the introduction. A clear description of questions such as why a 30-minute nap, why a RAST performance, and why a kickboxer should be added, and the uniqueness of this study should also be clearly described.’’
Correction 1: ‘‘ Revised. Sleep is a significant component known to severely alter or hinder performance outcomes [16]. Athletes, who are frequently exposed to high-intensity training and competition programs due to the physiological and psychological restorative effects of sleep, may need more sleep than the general population due to increased mental and physical requirements [16]. Consequently, daytime napping has been utilised as a strategy to increase the quality and quantity of sleep in athletes [17]. It has been reported that napping are crucial for supporting nighttime sleep and optimizing physical performance, as the recommended 8 hours of sleep per day may not be sufficient for athletes due to their physical and mental demands [16,18–23]. In addition napping opportunity can positively affect short-term maximal performance, attention, feelings, muscle soreness, fatigue, stress ((no-nap opportunity (N0), 25 min of nap opportunity (N25)) [20], vigilance, shuttle run (N25) [24], 5-m Shuttle Run Test ((no-nap opportunity (N0), 25 min of nap opportunity (N25), a 35 min nap opportunity (N35), a 45 min nap opportunity (N45)) (a 45 min nap opportunity (N45), a 90 min nap opportunity (N90)) [21,25], repeated sprint (a 20 min nap opportunity (N20), a 90 min nap opportunity (N90)) [26], sprint performance [27].
When all of these studies in the literature were analysed, it was determined that several of them investigated anaerobic performance. However, there are no studies evaluating kickboxers' anaerobic performance following the opportunity to lie down. During kickboxing competitions, rounds are often finished within two minutes, and anaerobic energy methods are typically utilised. RAST measures anaerobic capacity and anaerobic power [28]. Because of this, kickboxers were used in our study, and it was also intended to bring something new to the body of knowledge. In addition, although further research is required to determine the impacts of various nap protocols on the performance of kickboxers, the period during and after Ramadan appears to be a significant aspect in determining the degree of such an effect. This design can aid coaches in identifying the ideal scenario and enhancing the competitive preparation of kickboxers via a nap. This study aimed to assess the effects of 30 minutes of nap (N30) and AR on kickboxers' RAST performance. ’’
Comment 2: ‘‘IJERPH requires pre-registration for clinical trial registration. The presence or absence of clinical trial registration should be stated.’’
Correction 2: Necessary permissions were obtained from the ethics committee of Inonu University before the study was conducted, and the ethics approval number was stated in the manuscript.
Comment 3: ‘‘I don't think Figure 1 is information that should be included in the diagram.’’
Correction 3: Thank you. Figure 1 has been removed from the manuscript.

Round 2
Reviewer 2 Report
I felt that this manuscript was considerably improved compared to the previous one. At the same time, it is also a finding that has already been confirmed to be effective in previous research, and it is questionable whether it is unique enough to advance the knowledge in this field by applying it to "before and after Ramadan" and "anaerobic performance". . In addition, in recent years, in clinical intervention research, advance clinical trial registration has become essential, apart from the approval of the ethics committee. Therefore, there is no reason to think that this paper should be positively accepted.
Author Response
‘‘A Thirty Minute Nap Enhances Performance in Running Based Anaerobic Sprint Test During and After Ramadan Observance’’
Comments & Corrections
Reviewer 2 Comments
Comment 1: “I felt that this manuscript was considerably improved compared to the previous one.”
Correction 1: Thank you.
Comment 2: ‘‘At the same time, it is also a finding that has already been confirmed to be effective in previous research, and it is questionable whether it is unique enough to advance the knowledge in this field by applying it to "before and after Ramadan" and "anaerobic performance."
Correction 2: Thank you. Most of the studies in the literature investigate anaerobic performance between Ramadan periods. However, there are no studies evaluating kickboxers' anaerobic performance following the opportunity to nap. During kickboxing competitions, rounds are often finished within two minutes, and anaerobic energy methods are typically utilized. RAST measures anaerobic capacity and anaerobic power [28]. Because of this, kickboxers were used in our study, and it was also intended to bring something new to the body of knowledge. In addition, although further research is required to determine the impacts of various nap protocols on the performance of kickboxers, the period during and after Ramadan appears to be a significant aspect in determining the degree of such an effect. This design can aid coaches in identifying the ideal scenario and enhancing the competitive preparation of kickboxers via a NAP. As a result, this study was examined in two ways (Ramadan periods and the NAP factor). We think that this study is original with these aspects.
Comment 3: ‘‘In addition, in recent years, in clinical intervention research, advance clinical trial registration has become essential, apart from the approval of the ethics committee. Therefore, there is no reason to think that this paper should be positively accepted’’
Correction 3: Thank you. Non-Invasive Clinical Research is research that does not require direct human intervention. Studies on body physiology, such as exercise or studies based on anthropometric measurements are examples of Non-Invasive Clinical Trials. This study was approved by the Inonu University non-interventional clinical research ethics committee with the number 2022/3534. For this reason, clinical trial registration was not performed.
